Ultra-rapid detection of nuclear protein of severe fever with thrombocytopenia syndrome virus by colloidal gold immunochromatography assay

Huang Zhiwei 1 2
Li Jianhua 3
Wei Wentao 4
Li Hongyu 2
Yan Hao 3
Chen Ruyi 2
Li Jiaxuan 2
Tie Xiaotian 5
Wang Di 2
Wu Guangshang 2
Zhang Ling 2
Zhang Yanjun 3
Chen Keda 2 chenkd@zjsru.edu.cn
Lou Yongliang 1 lyl@wmu.edu.cn
1 School of Laboratory Medicine and Life Sciences, Wenzhou Medical University , Wenzhou , China
2 Key Laboratory of Artificial Organs and Computational Medicine in Zhejiang Province, Shulan International Medical College, Zhejiang Shuren University , Hangzhou , China
3 Zhejiang Provincial Center for Disease Control and Prevention , Hangzhou , China
4 Assure Tech. (Hangzhou) Co., Ltd. , Hangzhou , China
5 Zhejiang Chinese Medical University , Hangzhou , China
Grande-Pérez Ana
Electronic publication date: 2024 Oct 14
Publication date: 2024
Volume: 12
Electronic Location ID: e18275
Received 2024 Apr 30; Accepted 2024 Sep 18
Copyright: © 2024 Huang et al.
Copyright year: 2024
Copyright holder: Huang et al.
License: This is an open access article distributed under the terms of the Creative Commons Attribution License, which permits unrestricted use, distribution, reproduction and adaptation in any medium and for any purpose provided that it is properly attributed. For attribution, the original author(s), title, publication source (PeerJ) and either DOI or URL of the article must be cited.
License URL: https://creativecommons.org/licenses/by/4.0/

Keywords: Severe fever with thrombocytopenia syndrome virus, Monoclonal antibodies, Immunochromatography assay, Colloidal gold

Funding: Key Research and Development Program of Zhejiang Province 021C03044 Major Horizontal Project of Zhejiang Shuren University 2021D1034 National Key R&D Program of China 21YFC2301200 The Medical and Health Research Project of Zhejiang Province 2022KY127 Zhejiang Shuren University Basic Scientific Research 2024XZ014 This study was supported by the Key Research and Development Program of Zhejiang Province (Program Number 2021C03044), the Major horizontal project of Zhejiang Shuren University (2021D1034), the National key R&D Program of China (Projects Number 2021YFC2301200), the Medical and Health Research Project of Zhejiang Province (2022KY127) and the Zhejiang Shuren University Basic Scientific Research Special Funds (2024XZ014). The funders had no role in study design, data collection and analysis, decision to publish, or preparation of the manuscript.

==============================
In 2009, severe fever with thrombocytopenia syndrome virus (SFTSV), also known as the Dabie bandavirus (DBV), was first discovered in Henan, China. It is a tick-borne zoonotic virus with a fatality rate ranging from 6% to 30%. Currently, we lack safe and effective vaccines or antiviral drugs to treat SFTSV infection. Therefore, the development of a specific, sensitive, and cost-effective detection method is crucial. Using inactivated SFTSV and recombinant SFTSV nucleocapsid protein (SFTSV-NP), we repeatedly immunized mice with different adjuvants and obtained two monoclonal antibodies against SFTSV-NP, which were used to develop a colloidal gold immunochromatographic assay (ICA) rapid test kit for SFTSV. Compared with nucleic acid testing (gold standard), the ICA test strips are 97.67% accurate in testing clinical serum samples (36 cases of clinical serum samples and seven cases of whole blood samples). The test kit was 100% accurate in detecting different SFTSV strains. No false-positive results were generated when detecting other arboviruses. Therefore, our developed SFTSV test kit conveniently, rapidly, and effectively detects SFTSV.

Introduction

In 2009, Yu et al. (2011) first discovered and isolated a novel phlebovirus in Henan, China, subsequently named severe fever with thrombocytopenia syndrome virus (SFTSV). Although the International Committee on Taxonomy of Viruses (ICTV) formally classified the virus as Dabie bandavirus (DBV), the widely adopted term remains SFTSV; therefore, in this study, we continue to use SFTSV as the virus name (Casel, Park & Choi, 2021). The clinical symptoms of SFTSV infection primarily include acute fever, decreased platelet and white blood cell counts, gastrointestinal symptoms, and muscle pain. Severe cases might result in multi-organ complications and even death caused by multiple organ failure (Wang et al., 2022). Considering the non-specific clinical symptoms of SFTSV infection, relying solely on clinical symptoms is ineffective to distinguish SFTSV from other Bunyaviridae infections. Moreover, the mortality rate of SFTS ranges from 6% to 30%, which requires increased research attention. Despite ongoing research on the pathogenic mechanisms of SFTSV, including the establishment of various animal models, including murine models (Matsuno et al., 2017), ferret models (Park et al., 2019), and non-human primate models (Jin et al., 2015), the complex interactions between SFTSV and the host immune system remain incompletely understood. Considering that the population affected by SFTSV predominantly comprises permanent residents in mountainous and hilly regions with limited medical resources, there is a pressing need to develop an accurate, rapid, convenient, and cost-effective method to detect SFTSV.

SFTSV particles are spherical, enveloped, single-stranded RNA viruses with a diameter of 80 to 100 nm. Its genome consists of three single-stranded negative-sense RNA segments: L, M, and S. Among them, the M segment encodes the precursor membrane protein, which is cleaved into glycoproteins Gn and Gc. Gn binds to the cell surface non-muscle myosin heavy chain, facilitating the early infection phase of the virus. The S segment encodes the nucleocapsid protein (NP) and non-structural protein (Ns). The NP protein packages the viral RNA into ribonucleoprotein complexes (RNPs), preventing degradation by exogenous RNA nucleases or the host’s immune system. The L segment encodes the RNA-dependent RNA polymerase, which promotes the replication and transcription of the viral RNA (Li et al., 2021).

Currently, the detection methods for SFTSV primarily include virus isolation and identification, enzyme-linked immunosorbent assay (ELISA), quantitative real-time reverse transcription-polymerase chain reaction (qRT-PCR), and the clustered regularly interspaced short palindromic repeats (CRISPR)-CRISPR-associated protein (Cas) system detection (Umeki et al., 2020; Zhou et al., 2020; Zhu et al., 2022). Molecular detection methods require specialized laboratory personnel, and dedicated experimental facilities and equipment, making them unsuitable for regions with less advanced medical technology. In contrast, an immunochromatographic assay (ICA) is characterized by its simplicity, requiring no specialized personnel, and it offers a quick detection turnaround, typically displaying the results within 15–20 min. This makes it highly suitable for viral screening in epidemic areas. Colloidal gold-based immunochromatographic technology has been widely employed to detect various infectious diseases, such as SARS-CoV-2, influenza virus, and respiratory syncytial virus, among other (Wang et al., 2020; Yao et al., 2023). Zuo et al. (2018) developed test strips for the detection of SFTSV-NP protein by optimizing the conditions of article-based lateral flow immunochromatography test strips. Hence, in the present study, we attempted to apply colloidal gold immunochromatographic detection technology to SFTSV NP antigen detection (Fig. 1A). The detection principle involves the migration of the sample along the cellulose membrane towards the absorbent pad under chromatographic action. If the test sample contains the target antigen, it binds to the monoclonal antibody labeled with colloidal gold on the gold pad. Subsequently, it is captured by the pre-immobilized monoclonal antibody on the detection line, forming a visibly distinct red reaction line (Fig. 1B).

Figure 1 Description of the colloidal gold immunochromatography assay strip.

(A) The design and principle of the colloidal gold ICA strip. The sample to be tested can be either whole blood or serum. The sample to be tested moves through the NC membrane in the direction of the absorbent pad, passes through the colloidal conjugate pad, carries the colloidal gold-labeled antibody, and continues to move, which allows it to bind to the solid-phase antibody on the detection line (T) and the quality control line (C), respectively, and produces an obvious red line visible to the naked eye. (B) Rules for interpretation of ICA test strip results: The quality control line is an important indicator to evaluate the validity of the test, if the quality control line does not appear red, the test is invalid. Only after the quality control line and the detection line appear, is the test deemed positive, otherwise it is negative. (C) Color cards to interpret the ICA test strip results. IgG, immunoglobulin G; ICA, immunochromatographic strip; NC, negative control; SFTSV-NP, Severe Fever with Thrombocytopenia Syndrome Virus nucleocapsid protein; (D) Interpretation of the repetitive alternating immunization approach: Candidate adjuvants were mixed with immunogens. Inactivated virus cultures + recombinant SFTSV-NP protein were used for alternate immunization at 2-week intervals to obtain high-titer mice that met the fusion requirements. Created in BioRender (BioRender.com/d81d819).

Monoclonal antibodies, with their extremely high specificity and sensitivity, are widely applied in disease treatment and diagnosis. The dual-antibody sandwich immunoassay technique relies on monoclonal antibodies with high sensitivity and specificity to accurately determine the presence of the target antigen in samples. Hybridoma technology is currently the most widely used method to produce antibodies of non-human origin. It can be employed to generate monoclonal antibodies for almost any desired antigen. However, research has also indicated that the specificity of antibodies obtained through traditional hybridoma technology might be unsatisfactory (Mokhtary et al., 2022). Herein, a repetitive alternating immunization approach using a combination of recombinant proteins and inactivated virus cultures was employed for mouse immunization (Fig. 1D) was employed to obtain monoclonal antibodies with increased specificity and sensitivity.

Our team has previously explored the application of a peptide hydrogel adjuvant named SupraAjuvant in the preparation of H7N9 infection vaccines (Chen et al., 2023). Research indicates that hydrogel-adjuvanted vaccines can induce strong immune responses in mice, effectively protecting the host from H7N9 virus infection. Therefore, in this study, we also employed a novel peptide hydrogel adjuvant and compared its immunogenic effects with other commonly used adjuvants to investigate the differences in immune responses produced by different adjuvants.

In summary, to address the challenge of SFTSV detection in regions with limited medical resources, and to overcome the high cost associated with qRT-PCR that are only applicable to high-risk epidemic areas, we have developed the SFTSV-NP colloidal gold ICA kit by integrating novel adjuvant and immunological approach. This diagnostic kit demonstrates exceptional accuracy and specificity in detection. Given the absence of commercially available SFTSV-NP antigen detection methods, our groundbreaking SFTSV-NP ICA kit stands as an effective solution to the previously mentioned challenges.

Materials and Methods

Animals

The experimental mice, aged 7 weeks, were all Balb/C mice and purchased from Charles River Co., Ltd (Beijing, China). Mice were maintained and bred in specific pathogen-free conditions at Zhejiang Provincial Center for Disease Control and Prevention. Animal Ethics Committee of Zhejiang Provincial Center for Disease Control and Prevention provided full approval for this research (approval number: 2023-002-01).

Virus, cells, and clinical samples

SFTSV strain (SFTSV-2018-ZJ10), Chikungunya virus (Chik Vero 2P 12.8.6), and Zika virus (KU866423 | China | 2016) were preserved in Zhejiang Provincial Center for Disease Control and Prevention. Vero cells (ATCC CCL-81; Sinovac Biotech, Beijing, China) were cultured in minimum essential medium (MEM; Gibco, Grand Island, NY, USA). The Vero cell medium was supplemented with 10% fetal bovine serum (Gibco, Grand Island, NY, USA), 1% penicillin-streptomycin, 25 mM HEPES, and 2 mM L-glutamine and all cells were passaged every 2–3 days using trypsin-EDTA (0.25%, Gibco, Grand Island, NY, USA). Human sera from clinical samples were provided by Zhejiang Provincial Center for Disease Control and Prevention.This study was approved by the Ethical Review Committee of Zhejiang Provincial Center for Disease Control and Prevention, China (2023-002-01).

Main materials

Anti-His IgG, anti-Mouse IgG, and Goat Anti-Mouse IgG H&L (Alexa Fluor® 488) were purchased from Abcam (Cambridge, MA, USA). The mice used produce the monoclonal antibodies from hybridomas were purchased from Charles River Laboratories (Wilmington, MA, USA). The E. coli competent cells were purchased from Shanghai Weidi Biotechnology Co., Ltd. (Shanghai, China). QuickAntibody-Mouse and Freund’s adjuvant were purchased from Biodragon (Douglasville, GA, USA). SupraAjuvant was given by Zhimou, Yang (College of Life Sciences, Synergetic Innovation Center of Chemical Science and Engineering, and National Institute of Functional Materials, Nankai University).

Preparation of recombinant immunogens

The sequence information for the SFTSV-NP protein was obtained from NCBI. The sequence was optimized for expression in E. coli and synthesized by GenScript (Nanjing, China). Primers were designed to clone the target sequence into the pET-30a(+) expression vector (Primer F: 5′-GGGAATTCCATATGATGAGTGAATGGTCAAGGATAGCTG-3′ (Nde I); Primer R: 5′-AAATATCTCGAGCAGGTTGCGGTATGCA-3′ (Xho I). The ligation product was transformed into competent E. coli DH5α cells using heat shock. The recombinant plasmids were then transferred into the expression strain BL21(DE3). Expression of the SFTSV-NP recombinant protein was induced by adding isopropyl β-D-thiogalactoside (IPTG) at a concentration of 200 μg/mL at 25 °C with agitation. The bacterial cells were harvested, and the SFTSV-NP protein was purified using Ni2+ NTA resin affinity chromatography after cell lysis by ultrasonication at 200 rpm for 5 h.

The bacterial cells were resuspended in Binding Buffer (10 mM KH₂PO₄, 20 mM Na₂HPO₄·2H₂O, 500 mM NaCl, 50 mM imidazole, pH 8.0) and lysed using an ultrasonic cell disruptor (SCIENTZ-IID, Scientz, Ningbo, China). After sonication, the lysate was centrifuged at 12,000–13,000 rpm for 10 min at 4 °C. The supernatant was collected, filtered through a 1.2 μm filter using a sterile syringe, and set aside for purification.

A Ni-NTA affinity chromatography column was equilibrated with 10 column volumes of Binding Buffer. The sample was loaded onto the column at a flow rate of 2–4 mL/min (with a peristaltic pump at 3–6 rpm). The column was washed with Washing Buffer (10 mM KH₂PO₄, 20 mM Na₂HPO₄·2H₂O, 500 mM NaCl, 100 mM imidazole, pH 8.0) at 3–5 mL/min until the absorbance stabilized. The target protein was eluted using Elution Buffer (10 mM KH₂PO₄, 20 mM Na₂HPO₄·2H₂O, 500 mM NaCl, 500 mM imidazole, pH 8.0), and the fractions containing the protein were collected at the same flow rate. After use, the Ni-NTA column was washed with Washing Buffer, deionized water, and 20% ethanol before storage at 2–8 °C.

The eluate from the Ni-NTA column was dialyzed against 20 mM phosphate buffer (pH 7.4) and subjected to further purification using a DEAE column to optimize purity. The eluate from the DEAE column was dialyzed against phosphate buffer, sterile-filtered, sampled, and the final product was labeled and stored at −20 °C.

Preparation of monoclonal antibodies using hybridoma technology

Three adjuvants were chosen for this experiment: SupraAjuvant, QuickAntibody-Mouse, and Freund’s adjuvant. Ten mice were randomly assigned to each group. In this study, 7-week-old Balb/C mice were selected. Mouse immunization was performed using inactivated SFTSV cultures cross-immunized with recombinant SFTSV-NP, using four mice per round of each immunization method, at 2-week intervals. Immunized mice with antibody titers meeting the fusion requirements were selected for cell fusion using the PEG-mediated method, and semi-solid, limited dilution liquid culture was used to select the fusion cells. ELISA was used to screen the fusion cells. After obtaining positive clones, the clone was expanded to obtain the monoclonal antibodies. All mice were euthanized by cervical dislocation, whose spleen was taken for cell fusion. The two monoclonal antibody strains obtained were named SV01-13 and SV06-21.

Indirect enzyme-linked immunosorbent assay

After the fourth immunization, the titer of mouse serum was determined using an indirect ELISA. The recombinant SFTSV-NP protein was diluted to a concentration of 4 μg/mL. A volume of 100 μL of the diluted protein was added to each well of a 96-well plate, followed by incubation at 37 °C for 2 h in a constant temperature incubator. The coating solution was discarded, and the plate was washed five times with PBST (cwBio, Jiangsu, China). Following the washes, 200 μL of blocking buffer (cwBio, Jiangsu, China) was added to each well, and the plate was incubated at 37 °C for 1 h. The blocking buffer was then discarded, and the plate was washed five times with PBST. A total of 100 μL of diluted serum was added to each well, followed by incubation at 37 °C for 1 h. The liquid was discarded, the plate was washed five times with PBST, and 100 μL of HRP-labeled secondary antibody was added to each well. The plate was incubated at 37 °C for 1 h. After discarding the liquid and washing the plate five times with PBST, 50 μL of TMB (Solarbio Science & Technology Co., Ltd., Beijing, China) substrate was added to each well, followed by incubation at 37 °C for 10 min. The color development was stopped by adding 50 μL of stop solution (Solarbio Science & Technology Co., Ltd., Beijing, China) to each well. The OD450 value was then read using a SpectraMax Plus 384 microplate reader (Molecular Devices, Shanghai, China).

Western blot

We infected Vero cells with SFTSV virus, and after the cells showed lesions, samples of virus-infected cells were collected and lysed. The cell lysate was separated with sodium dodecyl sulfate–polyacry-lamide gel electrophoresis (SDS-PAGE), then transferred to polyvinylidene fluoride (PVDF) membranes. The blots were visualized with Western blot imaging equipment (Bio-Rad, Hercules, CA, USA). Briefly, the membranes were transferred to TBST (cwBio, Jiangsu, China) with 5% non-fat dry milk powder and shaken on a decolorizing shaker at room temperature for 1 h. SV01-13 and SV06-21 were diluted with antibody dilution buffer at 1:10,000 as primary antibodies. The membranes were removed from the sealing solution and incubated with the primary antibody at room temperature. The primary antibodies were shaken and incubated overnight on a decolorizing shaker at 4 °C. The blot was rinsed with TBST solution and incubated for 2 h with the secondary antibody (Vazyme, Nanjin China). After rinsing, the blots were incubated for 1 min with a LumiBest ECL Substrate solution kit, then observed in an imager.

Indirect immunofluorescence assay

Vero cells (3 × 106/well) were seeded into 12-well plates. After 48 h, the cells were infected with SFTSV. After 48 h, the medium was aspirated, the cells were washed once with PBS containing 2% BSA, and fixed for 15 min with methanol that had been pre-chilled for 15 min at −20 °C. After three washes with 2% BSA in PBS, the cells were permeabilized with 0.5% Triton X-100 (Solarbio Science & Technology Co., Ltd., Beijing, China) for 10 min. Then, the cells were blocked for 1 h with 2 mL 2% BSA, the blocking solution discarded, and the cells were washed three times with PBS. SV01-13 and SV06-21 were diluted with antibody dilution buffer at 1:10,000 as primary antibodies. Subsequently, 2 mL primary antibodies were added to each well separately and incubated at room temperature for 1 h or 4 °C overnight. Then, the sample was rinsed three times with 2% BSA for 5 min per rinse. Next, 2 mL secondary antibody was added to each well and incubated at room temperature for 1 h. Subsequently, the samples were rinsed three times with 2% BSA for 5 min per rinse. 4′,6-diamidino-2-phenylindole (DAPI) solution (2 mL, 1 mg/mL) was added to each well and reacted for 10 min in the dark. The analysis was observed with an EVOSTM M7000 imaging system.

Preparation of colloidal gold-recombinant antibodies

Colloidal gold particles were prepared by the Frans method (Wang et al., 2014). In 192 ml of ultrapure water with high-speed magnetic stirring, 8 ml of 1% chloroauric acid (HAuCl4) solution was added and boiled for 5 min, then 12.8 ml of 1% trisodium citrate solution was added. After boiling for 3 min, the colloidal gold pellet solution was cooled gradually. Bhosphate buffer (pH7.4) was added with an appropriate amount of monoclonal antibodies (SV01-13 and SV06-21) to a final concentration of 20 μg/mL, shaken and mixed well, added with 100 μL of prepared colloidal gold pellet solution, and shaken at 50–60 rpm at room temperature for 1 h. Thereafter, we added 20 μL of bovine serum albumin solution to block the reaction for 2 h, followed by centrifugation for 15 min at 500 × g. The supernatant was discarded, and the precipitate was resuspended by adding compound solution containing 1% BSA, 0.02% NaN3,5% trehalose and 20% sucrose.

Real-time reverse transcription PCR

We extracted RNA of SFTSV according to the TRIzol operating instructions (Invitrogen, Waltham, MA, USA). Total RNA was extracted using TRIzol and processed with RQ1 RNase-free DNase I (Promega, Madison, WI, USA) to eliminate residual DNA. DNA fragments were generated with the specific primers F (5′-AGCATGAATTCTCACGGAGC′) and R (5′-CGCTCTTCAAGGTTCTGCTT-3′) of the target gene (SFTSV-NP). According to the manufacturer’s instructions, the reaction was performed in a 25-μL reaction solution containing 17 μL of reaction solution A, 3 μL of reaction solution B and 5 μL of RNA elution. The RT-qPCR thermal cycling conditions were as follows: reverse transcription at 50 °C for 15 min and an initial denaturation step at 95 °C for 15 min, followed by 45 cycles of 94 °C for 15 s and 55 °C for 45 s. The amplification assay was performed on a 7500 Real-time PCR Detection System (Applied Biosystems, Waltham, MA, USA).

Evaluation of different strains of SFTSV

We used the developed ICA test to detect SFTSV strains preserved by the Zhejiang Provincial Center for Disease Control and Prevention, including SFTSV genotypes A, B, and D. The results of the ICA were interpreted with reference to the color card (Fig. 1C). Besides, we assessed the detection limits of different SFTSV strains, including dilutions and Ct values.The measurement sequence of all samples is randomly numbered.

Cross-reactivity of SFTSV detection

We collected nine clinical samples of dengue virus preserved by the Zhejiang Provincial Center for Disease Control and Prevention for testing, plus, Chikungunya virus and Zika virus, and evaluated the cross-reactivity capability of the test strip against various pathogens.

Detection of clinical samples

We utilized clinically suspected SFTSV infected serum samples preserved by the Zhejiang Provincial Center for Disease Control and Prevention. We dropped 75 μL of serum onto the ICA test strip to assess its detection capabilities for clinical specimens. Besides, considering that SFTSV mainly occurs in rural areas, in order to simplify the pre-processing procedures for blood samples, we tested whole blood samples from six healthy individuals and one positive case to evaluate the detection capability of the ICA test strip for whole blood samples.

Statistical methods

GraphPad Prism 8.0 software was used for graphical and statistical analyses.

Results

Preparation of recombinant immunogens

The recombinant pET-30a(+) plasmid expressing his-tagged SFTSV-NP was transformed into the Escherichia coli expression strain BL21 (DE3). After induction, the protein was purified using an Ni2+-NTA resin column. We separated the proteins using sodium dodecyl sulfate-polyacrylamide gel electrophoresis (SDS-PAGE) experiments and stained them with Coomassie Brilliant Blue R-250, which showed a single band corresponding to his-tagged SFTSV-NP (Fig. 2A). Western blotting was performed using an anti-his tag antibody (Fig. 2B), which indicated successful expression of the recombinant SFTSV-NP.

Figure 2 SFTSV-NP antigen and monoclonal antibody performance test results.

(A) SDS-PAGE results of the induction of recombinant SFTSV-NP protein expression and its purification: M, Marker; 1, Pre-purification sample; 2, SFTSV-NP protein samples purified using His tag-Ni columns; 3, SFTSV-NP samples decontaminated with 50 mM imidazole solution; 4, SFTSV-NP protein eluted with 500 mM imidazole solution. (B) Western blotting results of His-tag SFTSV-NP using anti-SFTSV-NP monoclonal antibodies SV01-13 and SV06-21. (C) Experimental results of indirect immunofluorescence assay using SV01-13 and SV06-21 as the primary antibodies after natural virus infection of Vero cells. SDS-PAGE, sodium dodecyl sulfate-polyacrylamide gel electrophoresis; SFTSV-NP, severe fever with thrombocytopenia syndrome virus nucleocapsid protein; DAPI, 4¢,6-diamidino-2-phenylindole.

Immunization of mice with different adjuvants

In this study, we employed three different adjuvant immunization schemes (SupraAjuvant, QuickAntibody-Mouse and Freund’s adjuvant). The comparative analysis revealed that the SupraAjuvant yielded the most effective immune titers, followed by QuickAntibody-Mouse, and lastly Freund’s adjuvant. Thus, it was evident that the peptide hydrogel adjuvant could indeed stimulate the host to generate a stronger immune response (Fig. 3A).

Figure 3 Mouse immunization results and antibody performance validation.

(A) Results of antibody titers in different mice (M8-T, M8-M, M8-W, M8-ZQ) using indirect ELISA after immunization with three different adjuvant immunization protocols. (B) Results of antibody potency testing for anti-SFTSV-NP monoclonal antibodies SV01-13 and SV06-21. (C) Results of ICA test strips for SFTSV clinical serum sample testing.

Preparation and validation of anti-SFTSV-NP monoclonal antibodies

After obtaining multiple positive clones by the hybridoma technique, two monoclonal antibodies, SV01-13 and SV06-21, were obtained by screening two monoclonal antibodies with high affinity for SFTSV. We performed performance validation on both of the obtained monoclonal antibodies, SV01-13 and SV06-21. SV01-13 and SV06-21 possessed good potency, and the working concentration could reach 15.625 ng/ml, as detected using indirect ELISA (Fig. 3B). Vero cells were then cultured and infected with the SFTSV virus. After the appearance of a cytopathic effect, the cells were collected. Western blotting experiments were conducted using SV01-13 and SV06-21 as the primary antibodies at a dilution of 1:12,000, with anti-mouse IgG as the secondary antibody (Fig. 2B). Similarly, Vero cells were infected with the SFTSV virus and fixed after the appearance of the cytopathic effect. Indirect immunofluorescence assays were performed using SV01-13 and SV06-21 as the primary antibodies at a dilution of 1:10,000, and fluorescence-labeled anti-mouse IgG as the secondary antibody (Fig. 2C). The experimental results showed that both monoclonal antibodies could specifically bind to the NP protein of the SFTSV virus strain, demonstrating excellent sensitivity.

Evaluation of different strains of SFTSV

Through testing different strains of SFTSV, the results indicated a 100% detection rate for the immunochromatographic strip (ICA) test for the various SFTSV strains. In comparison with the qRT-PCR results, it was evident that the average detection limit of this assay kit was a Ct value of 30.142, demonstrating excellent detection capabilities (Fig. 4A and Table 1).

Figure 4 The results of SFTSV ICA.

(A) The results of different strains of SFTSV and cross-reactivity of DENV, ZIKV and CHIKV. The results showed that ICA was able to accurately detect different SFTSV strains and was negative for cross-testing DENV, ZIKV and CHIKV. (B) The results of ICA test in clinical serum samples. In the analysis of 36 serum samples, 30 were determined to be positive, and six were negative (SFTSV-25, 31, 44, 52, 54, 57) according to RT-qPCR results. The ICA approach accurately identified 29 positive samples out of the 30 confirmed positives. Notably, one specimen, SFTSV-9, was incorrectly classified as negative, with a Ct value of 34.86 falling within the ICA detection error range; (C) The results of whole blood samples. All seven whole blood samples were accurate (six negative samples and one positive sample) for colloidal gold ICA testing.

Table 1 Detection results of different strains of SFTSV and Ct values at the limit of detection.

SFTSV strain number	Genotype of the L fragment	Genotype of the M fragment	Genotype of the S fragment	Virus titer/ TCID50/mL	Ct value at the limit of detection	Ct mean	ICA test results	
SFTSV-8	D	D	D	106.7	31.429	30.142	All positive	
SFTSV-9	A	A	A	105.7	28.081	
SFTSV-28	B	B	B	107.1	31.751	
SFTSV-36	B	B	B	105.9	29.435	
SFTSV-39	D	B	B	106.8	30.810	
SFTSV-60	B	B	B	106.5	30.114	
SFTSV-105	D	D	D	105.9	29.375	

Cross-reactivity of SFTSV detection

We collected nine clinical serum samples of Dengue virus preserved by the Zhejiang Provincial Center for Disease Control and Prevention, and single samples of Zika Virus and Chikungunya Virus, to evaluate the cross-detection capability of the test strip against various pathogens (Fig. 4A and Table 2). None of the 11 samples tested positive.

Table 2 Detection results of ICA test strip cross-reactivity.

Virus name	Virus type	Ct values	ICA test results	
Dengue virus	DENV-1	32.19	All negative	
DENV-1	23.19	
DENV-1	19.3	
DENV-2	34	
DENV-2	29.18	
DENV-2	25.79	
DENV-2	15.25	
DENV-3	27	
DENV-3	29	
Zika virus	Asian genotype	21	Negative	
Chikungunya virus	Asian genotype	19	Negative	

Detection of clinical samples

We analyzed 36 clinical serum samples and compared the results with those of qRT-PCR detection (Fig. 4B). We observed that among the 36 clinical serum samples, the positive detection rate of qRT-PCR-positive samples (30 cases) with Ct values in the range of 15–35 was 96.67% (29 cases), and the negative detection rate of qRT-PCR-negative samples (seven cases) with Ct values ≥ 35 and was 85.71% (six cases) (Fig. 4B). The only serum sample with a false-negative ICA test result had a Ct value of 34.86, which is in the error interval of the ICA test (Fig. 3C and Table 3). Additionally, the colloidal gold kit ICA test was 100% correct for whole blood samples, correctly detecting six negative samples and one positive sample (Fig. 4C).

Table 3 Detection of clinical serum samples and corresponding Ct values.

Ct value range	Clinical sample cases	Number of positive ICA test results	Number of negative ICA test results	
15 < Ct ≤ 20	1 (2.78%)	1 (2.78%)	0	
20 < Ct ≤ 25	0	0	0	
25 < Ct ≤ 30	8 (22.22%)	8 (22.22%)	0	
30 < Ct ≤ 35	21 (58.33%)	20 (55.56%)	1 (2.78%)	
35 < Ct ≤ 40	6 (16.67%)	0	6 (16.67%)	
Total	36	29 (80.56%)	7 (19.44%)	

Discussion

We obtained a pair of anti-SFTSV-NP monoclonal antibodies with high affinity and specificity using an alternate repeat immunization protocol of “recombinant protein + inactivated virus” and a novel hydrogel adjuvant. This pair of monoclonal antibodies was combined with colloidal gold detection technology to develop a colloidal gold rapid diagnostic test kit via an immunochromatography assay to detect the SFTSV-NP antigen. The detection of different strains of SFTSV indicated that the positive compliance rate of the kit for different strains of SFTSV reached 100%. The cross-reactivity-tests showed that the method was highly specific to detect the SFTSV-NP protein, and no cross-reactivity was detected for Dengue virus, Zika virus, and Chikungunya virus. Using clinical serum samples, the ICA strip identified a total of 29 positive samples, one false-negative sample, and six negative samples among 30 positive serum samples and six negative serum samples (compared with the results of the qRT-PCR assay). The false-negative sample exhibited a Ct value of 34.86, falling within the error range of the ICA detection. Besides, for the detection of whole blood samples, the ICA test strips also demonstrated commendable detection capability, with all six whole blood samples yielding negative results and one positive result. It should be noted that this positive whole blood sample underwent severe hemolysis during the blood collection process, which explains the higher background in the ICA test result. Nevertheless, we were still able to observe a clear red positive test line in the ICA result interpretation. SFTSV is currently predominantly occurring in rural areas such as mountainous regions. Experimental validation of whole blood sample detection confirms that the ICA test strips can be utilized for self-testing in medically resource-limited areas. The test strips can be applied for detection without the need for pre-processing of blood samples, significantly enhancing operational convenience. These results indicate that the SFTSV-NP colloidal gold immunochromatographic assay kit possesses excellent detection performance.

Currently, the detection of SFTSV is primarily based on PCR, quantitative reverse transcription PCR (qRT-PCR) (Sun et al., 2012), and loop-mediated isothermal amplification (LAMP) (Sano et al., 2021), all of which target SFTSV RNA for detection. In addition, emerging CRISPR systems have been developed for SFTSV detection. Apart from detecting the pathogen, these systems can also differentiate between different genotypes of SFTSV (Huang et al., 2022). Although these methods demonstrate better sensitivity and specificity compared to serological approaches, they pose significant challenges for experimental personnel and laboratory facilities, demanding a substantial amount of time. In fact, serological detection methods for SFTSV have also emerged, including immunochromatographic techniques and ELISA methods for SFTSV antibodies. Considering the abundant and stable nature of the NP antigen, these methods primarily use SFTSV-NP antibodies as targets for detection. Nevertheless, the window period for antibody detection often occurs around 2 weeks after exposure and may not be completed within a short time after SFTSV infection. Therefore, SFTSV-NP antigen detection may be a viable option worth exploring for point-of-care testing (POCT). Several colloidal gold assays have been applied to the rapid detection of SFTSV. Wang et al. (2014) achieved rapid detection of SFTSV by detecting IgG and IgM antibodies to SFTSV, while Zuo et al. (2018) determined whether SFTSV was infected or not by detecting SFTSV-NP proteins. Notably, Zuo et al. (2018) optimized the production conditions of colloidal gold test strips to enable ICA to achieve higher detection sensitivity. In the present study, the SFTSV-NP monoclonal antibodies with high detection performance were prepared by optimizing the immunization method and applying a novel adjuvant, and the detection kit was successfully prepared by applying the monoclonal antibodies.

However, this study still has some limitations. Firstly, the mouse immunization protocol in this study was not comprehensively compared with traditional immunization methods. It can only be inferred that the novel adjuvants and alternate immunization scheme resulted in better immune effects, but it does not prove that the novel immunization approach in this study is superior to traditional immunization methods in terms of immune efficacy. Secondly, the performance validation of the colloidal gold ICA test kit was not conducted with a large number of clinical samples; the sample size was insufficient. Especially, the number of positive whole blood samples is very low. It is recommended to supplement the sample size in subsequent optimization processes. Additionally, for the high-quality monoclonal antibodies obtained in this study, further antibody research is warranted, for example, determining antibody epitope regions.

Based on the current experimental foundation, in order to enhance both the yield and quality of antibodies, we will transition from the original antibody production method to eukaryotic expression to obtain these two antibodies (Tan et al., 2021). We are currently conducting sequencing of the two antibodies. Once sequencing is complete, we will use the HEK293 eukaryotic cell expression system to directly express antibodies SV01-13 and SV06-21. By utilizing the HEK293 eukaryotic cell expression system, antibody production can be significantly increased, facilitating the transition from laboratory research to industrial-scale production.

Conclusions

Overall, the method is capable of preparing monoclonal antibodies with high specificity and sensitivity. The colloidal gold rapid test diagnostic kit for the SFTSV-NP antigen can detect the pathogen rapidly and accurately, suggesting that it could be used for population screening in epidemic areas and for point-of-care testing.

Supplemental Information

Supplemental Information 1 Original Western Blots.

Supplemental Information 2 The sequence of SFTSV-NP.

Supplemental Information 3 Author Checklist.

We are very grateful to my supervisors Yanjun Zhang, Keda Chen and Yongliang Lou for their valuable and constructive advice while I was writing this article. We sincerely thank Zhimou Zhang (College of Life Sciences, Synergetic Innovation Center of Chemical Science and Engineering, and National Institute of Functional Materials, Nankai University) for providing SupraAjuvant. We sincerely thank Hao Yan (Zhejiang Provincial Center for Disease Control and Prevention, Hangzhou, China) for providing clinical samples and different strains of SFTSV, DENV, ZIKV and CHIKV. Figures were created using BioRender.com.

Additional Information and Declarations

Competing Interests

Author Contributions

Animal Ethics

Data Availability

Wentao Wei is employed by Assure Tech. (Hangzhou) Co., Ltd. The authors declare that they have no competing interests.

Zhiwei Huang performed the experiments, analyzed the data, prepared figures and/or tables, authored or reviewed drafts of the article, and approved the final draft.

Jianhua Li conceived and designed the experiments, performed the experiments, prepared figures and/or tables, and approved the final draft.

Wentao Wei conceived and designed the experiments, performed the experiments, prepared figures and/or tables, and approved the final draft.

Hongyu Li analyzed the data, authored or reviewed drafts of the article, and approved the final draft.

Hao Yan performed the experiments, authored or reviewed drafts of the article, and approved the final draft.

Ruyi Chen performed the experiments, authored or reviewed drafts of the article, and approved the final draft.

Jiaxuan Li analyzed the data, authored or reviewed drafts of the article, and approved the final draft.

Xiaotian Tie analyzed the data, prepared figures and/or tables, and approved the final draft.

Di Wang analyzed the data, prepared figures and/or tables, and approved the final draft.

Guangshang Wu analyzed the data, prepared figures and/or tables, and approved the final draft.

Ling Zhang analyzed the data, prepared figures and/or tables, and approved the final draft.

Yanjun Zhang conceived and designed the experiments, prepared figures and/or tables, authored or reviewed drafts of the article, and approved the final draft.

Keda Chen conceived and designed the experiments, prepared figures and/or tables, and approved the final draft.

Yongliang Lou conceived and designed the experiments, authored or reviewed drafts of the article, and approved the final draft.

The following information was supplied relating to ethical approvals (i.e., approving body and any reference numbers):

The Ethical Review Committee of Zhejiang Provincial Center for Disease Control and Prevention, China.

The following information was supplied regarding data availability:

The Western blots are available in the Supplemental Files.

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
