# Peer review of "Ultra-rapid detection of nuclear protein of severe fever with thrombocytopenia syndrome virus by colloidal gold immunochromatography assay"

_PeerJ, doi:10.7717/peerj.18275_

## Round 0.1 · original submission · Major Revisions

Your manuscript has raised the interest of three referees but they have concerns that need to be addressed. I believe all comments are very constructive and will help your work to be suitable for publication. Please carefully review all the issues to convincingly show the validity of your method and the improvement with respect to previously reported methods. Importantly, do not forget to check for English language throughout the manuscript and make sure methods are sufficiently detailed, this is a key point in your work. Finally, a thorough discussion will be great. I am sure it will really improve and make your work shine! I am looking forward to receiving your answers to the reviewers comments and the edited manuscript

Reviewer 1 ·

Basic reporting

The authors should cite as a reference the paper reported by Zuo et al (https://www.ncbi.nlm.nih.gov/pmc/articles/PMC6288773/). The paper addressed a rapid detection of SFTSV via colloidal gold immunochromatography assay. It seems that authors should indicate differences from the prior publication. This is mentioned neither in Introduction nor in Discussion.

Experimental design

no comment

Validity of the findings

no comment

Additional comments

The manuscript reports to the development of a colloidal gold immunochromatographic assay rapid test kit for SFTSV. The development of a rapid, simple, and low-cost SFTSV detection method and the detailed methodology report are greatly fine.
This paper presents very interesting data and methods, but it still needs an extensive revision to be Peer J. The existence of a similar report in the past is the most important consideration.

-Major comments
1. The authors obtained two monoclonal antibodies, SV01-13 and SV06-21. However, they didn’t state which monoclonal antibody was used for ICA. If both monoclonal antibodies were used, the proportions should be stated in Materials & Methods.

2. The authors don’t mention methods of antibody detection (ELISA, IFA and Western blotting) and real-time RT-PCR in Materials & Methods. The method of detection of antibodies should be clearly described because it is discussed that the novel immunization technique. Since the sensitivity and specificity of the ICA test are evaluated by comparison with real-time RT-PCR, the method of real-time RT-PCR should be clearly indicated.

3. Whenever possible, it should be indicated by RNA copy number or virus titer (e.g., PFU) in addition to Ct value. Alternatively, if the virus titer or number of copies of viral RNA are below the limit of detection, the authors should clearly indicate.

4. The prior report have shown the detection limits in terms of target protein concentration (Zuo et al.). The detection limit in the ICA should be determined using concentration of SFTSV nucleocapsid proteins for comparison with the prior report.

-Minor comments
Line 37 and 62: The authors describe “a fatality rate ranging from 6% to 30%” in line 37. On the other hand, they describe “the mortality rate of SFTS ranges from 6% to 27%” in line 62. The authors should unify their interpretations and clearly indicate references.

Line 161: From the context, it can be inferred that “Supra Ajuvant” is a peptide hydrogel adjuvant. I apologize if I missed it, I cannot find the mention of this. It is an important aspect of the development of the ICA, and the authors should be clearly state. If there is any report, a reference to "Supra Ajuvant" should also be indicated.
I apologize if I missed it,

Line 175: Please correct “added After” to “added. After”.

Line 184: The authors should clearly describe the strains of SFTSV used in this study, including the nucleotide sequence.

Line 231: Please correct “demonstrating excellent specificity” to “demonstrating excellent sensitivity”.

Line 259-260: It is not surprising that viruses of different families are not crossed. If these viruses require differential diagnosis in the field, the authors should be clearly indicated.

Figure. 4A: The maximum value of absorbance differs from graph to graph. This should be aligned. It is difficult to discuss these graphs because the ELISA method is not stated in the manuscript. Fuch's adjuvant is not mentioned in the manuscript. Please check this title.

·

Basic reporting

In the manuscript titled “Ultra-rapid detection of nuclear protein of Severe Fever with Thrombocytopenia Syndrome Virus by colloidal gold immunochromatography assay,” Huang et al. aim to develop a specific, sensitive, and cost-effective method to detect SFTSV.
Overall, the manuscript has clear research aims. However, the English language needs improvement for conciseness and readability, particularly in lines 105-108, 142-143, 150-159, and 233-234. Additionally, ensure that scientific names, such as "E. coli," are italicized, and be consistent with abbreviations (e.g., SFTSV vs. SFTS in lines 60 and 62). All manufacturers of commercial reagents and software should be mentioned.
The literature review is insufficient. For example, the method used by Zuo et al. in 2018 (https://doi.org/10.1021/acsomega.8b02366) is relevant but not cited.
There are redundancies between the Methods and Results sections. The results need to be better interpreted and reported.
To broaden the audience, it would be helpful to include an explanation of adjuvant immunization schemes and their importance in this study.
Figures:
• Sample replications within the research are missing.
• Figure 2A: The sample explanations are unclear. Please clarify the samples in each lane with more details in the Method section.
• Figure 3A: The colors need to be re-selected for better distinction between titers, as the two darker blues are almost identical.
Raw data:
Please include the sequences of antigen and primers used in the study.

Experimental design

Several experiments are unclear or missing details.
The section "Preparation of recombinant immunogens" needs more specificity. Please include detailed information on:
• Cloning: Please provide the sequences of the antigen and the primers used.
• Protein Expression: Please specify IPTG concentration, expression temperature and duration, as well as cell harvest and storage conditions.
• Purification: Please describe in details all buffer conditions, the type of Ni-NTA resin or column used, and cell disruption conditions.
Western Blot and ELISA methods should be described in detail in the Methods section, including all steps involved.

Validity of the findings

The research employs the same concept of detecting the SFTSV antigen using a Colloidal Gold Immunochromatography Assay, first published by Zuo et al. in 2018 (https://doi.org/10.1021/acsomega.8b02366). However, the authors did not cite this prior publication or discuss other relevant studies using the same method.
The authors used monoclonal antibodies (mAbs) from mice. Is this method suitable for the large-scale production of mAbs for quick test kits?
The sample numbers were low, and the experiments lacked replicates. I suggest the authors compare their results and methods with previous publications using the same idea, highlighting any improvements made in their research.

Additional comments

The manuscript requires major revisions before it can be published. Key issues include:
• A lack of thorough discussion.
• Unclear and insufficiently detailed methods.
• Weak structure in the Results section.
In its current form, the manuscript does not convincingly demonstrate that the developed methods are improvements over previous publications.

Reviewer 3 ·

Basic reporting

Huang et al research on Severe Fever with thrombocytopenia syndrome virus rapid detection assay development, is very important considering the recent outbreaks of other tick-borne viruses surging. SFTS, again pose the serious threat. The experiments were well performed and all the data was presented relevant to design.

Experimental design

The experiments were properly designed and address all the concerns. Th process of recombinant protein generation and purification was accurate. Also, the validation of SupraAdjuvant was a good start to look into immunoreactivity. Validation of the Antibodies using different techniques and controls are followed accurately

Validity of the findings

it would be helpful if authors could include the details for the requirements criteria, they chose for fusion using PEG-mediated method (in line 166) and include the ELISA results. Also, if they could explain the (257-258) the positive compliance rate of kit to be 100%, when SFTSV-33, 36, 37, 41, 49, 51, 55, 56, 58 looked negative.
Also, the positive whole blood sample has lot of background, if that could be explained and if possible, more positive controls for whole blood be included it will bolster the findings.

Additional comments

I appreciate the hard work and the findings the authors presented in their study, if the above concerns are addressed and improved, it shall be accepted

---

## Round 0.2 · Minor Revisions

Thank you for the revised manuscript. The changes you have made are generally well-executed and address most of the concerns raised in the previous review. However, some of the newly added text in the Materials and Methods section would benefit from further proofreading to correct minor grammatical errors and improve clarity.

For example, the following section:

"Preparation of Recombinant Immunogens
We obtained the sequence information of the SFTSV-NP protein from NCBI. The sequence was optimized for expression in E. coli and entrusted to GenScript (Nanjing) Co.Ltd. for whole-genome synthesis. Primer sequences were designed to clone the target sequence into the expression vector pET-30a(+). ( Primer F: 5’-GGGAATTCCATATGATGAGTGAATGGTCAAGGATAGCTG-3’ (Nde I), Primer R: 5’-AAATATCTCGAGCAGGTTGCGGTATGCA-3’ (Xho I)). The ligation product was transformed into competent E. coli DH5α cells using heat shock, and the large amount of recombinant plasmids obtained were transferred into the expression strain BL21(DE3). Isopropyl β-D-Thiogalactoside (cwBio, Jiangsu, China) was added at 200 μg/ml to induce the expression of SFTSV-NP recombinant protein under 25 ℃ and 200 rpm lasting 5 hours.
The bacterial cells obtained in the previous step are resuspended in Binding Buffer (10mM KH2PO4、20mM Na2HPO4·2H2O、500mM NaCl、50mM Imidazole,pH8.0), and then lysed using SCIENTZ-IID ultrasonic cell disruptor (Scientz, Ningbo, China). After sonication, the mixture is centrifuged at 12,000-13,000 rpm for 10 minutes at 4°C. The supernatant is collected, and the pellet is discarded. The supernatant, which is the sample to be purified, is filtered through a 1.2 μm filter using a sterile disposable syringe to remove any particulates and set aside.
Before purification, select an appropriately sized Ni-NTA affinity chromatography column, fill it with the resin, and check the chromatography setup. Once the column is ready, equilibrate the Ni-NTA column with 10 column volumes of Binding Buffer. The sample is then loaded onto the column at a flow rate of 2-4 mL/min (with a peristaltic pump speed of 3-6 rpm). Wash the column with Washing Buffer (10mM KH2PO4、20mM Na2HPO4·2H2O、500mM NaCl、100mM Imidazole,pH8.0) until the absorption peak stabilizes, at a flow rate of 3-5 mL/min. Wash the column with Washing Buffer again at the same flow rate. Elute the target protein using Elution Buffer (10mM KH2PO4、20mM Na2HPO4·2H2O、500mM NaCl、500mM Imidazole,pH8.0) and collect the eluted fractions containing the desired protein, again at a flow rate of 3-5 mL/min. After use, the Ni-NTA column should be washed with Washing Buffer, deionized water, and 20% ethanol, and then stored at 2-8°C, maintaining the same flow rate.
The eluate obtained from the Ni-NTA column is dialyzed into a 20 mM phosphate buffer (pH 7.4) and then subjected to a second round of purification using a DEAE column to further optimize purity. The eluate from the DEAE column is dialyzed into phosphate buffer, sterile-filtered, sampled, and the final product is labeled and stored at -20°C."


could be improved as follows:

"Preparation of Recombinant Immunogens

The sequence information for the SFTSV-NP protein was obtained from NCBI. The sequence was optimized for expression in E. coli and synthesized by GenScript (Nanjing, China). Primers were designed to clone the target sequence into the pET-30a(+) expression vector (Primer F: 5’-GGGAATTCCATATGATGAGTGAATGGTCAAGGATAGCTG-3’ (Nde I); Primer R: 5’-AAATATCTCGAGCAGGTTGCGGTATGCA-3’ (Xho I)). The ligation product was transformed into competent E. coli DH5α cells using heat shock. The recombinant plasmids were then transferred into the expression strain BL21(DE3). Expression of the SFTSV-NP recombinant protein was induced by adding isopropyl β-D-thiogalactoside (IPTG) at a concentration of 200 μg/mL at 25°C with agitation. The bacterial cells were harvested, and the SFTSV-NP protein was purified using Ni²⁺ NTA resin affinity chromatography after cell lysis by ultrasonication at 200 rpm for 5 hours.

The bacterial cells were resuspended in Binding Buffer (10 mM KH₂PO₄, 20 mM Na₂HPO₄·2H₂O, 500 mM NaCl, 50 mM imidazole, pH 8.0) and lysed using an ultrasonic cell disruptor (SCIENTZ-IID, Scientz, Ningbo, China). After sonication, the lysate was centrifuged at 12,000–13,000 rpm for 10 minutes at 4°C. The supernatant was collected, filtered through a 1.2 μm filter using a sterile syringe, and set aside for purification.

A Ni-NTA affinity chromatography column was equilibrated with 10 column volumes of Binding Buffer. The sample was loaded onto the column at a flow rate of 2–4 mL/min (with a peristaltic pump at 3–6 rpm). The column was washed with Washing Buffer (10 mM KH₂PO₄, 20 mM Na₂HPO₄·2H₂O, 500 mM NaCl, 100 mM imidazole, pH 8.0) at 3–5 mL/min until the absorbance stabilized. The target protein was eluted using Elution Buffer (10 mM KH₂PO₄, 20 mM Na₂HPO₄·2H₂O, 500 mM NaCl, 500 mM imidazole, pH 8.0), and the fractions containing the protein were collected at the same flow rate. After use, the Ni-NTA column was washed with Washing Buffer, deionized water, and 20% ethanol before storage at 2–8°C.

The eluate from the Ni-NTA column was dialyzed against 20 mM phosphate buffer (pH 7.4) and subjected to further purification using a DEAE column to optimize purity. The eluate from the DEAE column was dialyzed against phosphate buffer, sterile-filtered, sampled, and the final product was labeled and stored at -20°C."

Please review this section of the manuscript carefully to ensure similar consistency and clarity throughout. Once these adjustments are made, the manuscript should be ready for final acceptance.

Thank you for your continued efforts.

---

## Round 0.3 · Minor Revisions

Thank you for the changes; they look good overall. However, the paragraph titled "Indirect Enzyme-Linked Immunosorbent Assay" in the Materials and Methods section requires further revision. I have reviewed the text and adjusted the verb tenses for consistency, as the procedures should be described in the past tense. Additionally, I made minor rewording changes to reduce repetition and improve clarity. Please review the revised version and update the manuscript accordingly. Once this is done, I will be able to accept the manuscript.

Indirect Enzyme-Linked Immunosorbent Assay

After the fourth immunization, the titer of mouse serum was determined using an indirect ELISA. The recombinant SFTSV-NP protein was diluted to a concentration of 4 μg/mL. A volume of 100 μL of the diluted protein was added to each well of a 96-well plate, followed by incubation at 37°C for 2 hours in a constant temperature incubator. The coating solution was discarded, and the plate was washed five times with PBST (cwBio, Jiangsu, China). Following the washes, 200 μL of blocking buffer (cwBio, Jiangsu, China) was added to each well, and the plate was incubated at 37°C for 1 hour. The blocking buffer was then discarded, and the plate was washed five times with PBST. A total of 100 μL of diluted serum was added to each well, followed by incubation at 37°C for 1 hour. The liquid was discarded, the plate was washed five times with PBST, and 100 μL of HRP-labeled secondary antibody was added to each well. The plate was incubated at 37°C for 1 hour. After discarding the liquid and washing the plate five times with PBST, 50 μL of TMB (Solarbio Science & Technology Co., Ltd., Beijing, China) substrate was added to each well, followed by incubation at 37°C for 10 minutes. The color development was stopped by adding 50 μL of stop solution (Solarbio Science & Technology Co., Ltd., Beijing, China) to each well. The OD450 value was then read using a SpectraMax Plus 384 microplate reader (Molecular Devices, Shanghai, China).

---

## Round 0.4 · accepted · Accept

I am satisfied with the improvements made to the updated version. The manuscript has been sufficiently revised and is now ready for publication. Thank you.